# Characterization of a DCL2-Insensitive *Tomato Bushy Stunt Virus* Isolate Infecting *Arabidopsis thaliana*

**DOI:** 10.3390/v12101121

**Published:** 2020-10-02

**Authors:** Marco Incarbone, Hélene Scheer, Jean-Michel Hily, Lauriane Kuhn, Mathieu Erhardt, Patrice Dunoyer, Denise Altenbach, Christophe Ritzenthaler

**Affiliations:** 1Institut de Biologie de Moléculaire des Plantes, CNRS, Université de Strasbourg, 67000 Strasbourg, France; helene.scheer@ibmp-cnrs.unistra.fr (H.S.); mathieu.erhardt@ibmp-cnrs.unistra.fr (M.E.); patdunoyer@gmail.com (P.D.); 2IFV, Le Grau-Du-Roi, Université de Strasbourg, INRAE, SVQV UNR-A 1131, 68000 Colmar, France; jean-michel.hily@vignevin.com; 3Plateforme protéomique Strasbourg Esplanade FR1589 du CNRS, Université de Strasbourg, 67000 Strasbourg, France; l.kuhn@ibmc-cnrs.unistra.fr; 4Bioreba AG, Christoph Merian Ring 7, CH-4153 Reinach, Switzerland; altenbach@bioreba.ch

**Keywords:** *Tomato bushy stunt virus*, *Arabidopsis*

## Abstract

*Tomato bushy stunt virus* (TBSV), the type member of the genus *Tombusvirus* in the family *Tombusviridae* is one of the best studied plant viruses. The TBSV natural and experimental host range covers a wide spectrum of plants including agricultural crops, ornamentals, vegetables and *Nicotiana benthamiana*. However, *Arabidopsis thaliana*, the well-established model organism in plant biology, genetics and plant–microbe interactions is absent from the list of known TBSV host plant species. Most of our recent knowledge of the virus life cycle has emanated from studies in *Saccharomyces cerevisiae*, a surrogate host for TBSV that lacks crucial plant antiviral mechanisms such as RNA interference (RNAi). Here, we identified and characterized a TBSV isolate able to infect *Arabidopsis* with high efficiency. We demonstrated by confocal and 3D electron microscopy that in *Arabidopsis* TBSV-BS3Ng replicates in association with clustered peroxisomes in which numerous spherules are induced. A dsRNA-centered immunoprecipitation analysis allowed the identification of TBSV-associated host components including DRB2 and DRB4, which perfectly localized to replication sites, and NFD2 that accumulated in larger viral factories in which peroxisomes cluster. By challenging knock-out mutants for key RNAi factors, we showed that TBSV-BS3Ng undergoes a non-canonical RNAi defensive reaction. In fact, unlike other RNA viruses described, no 22nt TBSV-derived small RNA are detected in the absence of DCL4, indicating that this virus is DCL2-insensitive. The new *Arabidopsis*-TBSV-BS3Ng pathosystem should provide a valuable new model for dissecting plant–virus interactions in complement to *Saccharomyces cerevisiae*.

## 1. Introduction

*Tomato bushy stunt virus* (TBSV) is the type member of the genus *Tombusvirus* in the family *Tombusviridae*. First isolated in 1935 in Ireland [1], this soil-borne plant pathogen for which no biological vector is known, is readily transmitted by mechanical inoculation. Symptoms induced by TBSV are largely dependent on the plant host and vary from necrotic and chlorotic lesions, to a mild or severe mosaic, or even lethal systemic necrosis in extreme cases. TBSV has a host range limited in nature mostly to dicotyledonous species including agricultural crops, ornamentals and vegetables such as tomato, pepper, lettuce, tulip, cherry, apple and grapevine [2]. Despite this wide distribution, TBSV is not considered an economically significant plant pathogen. The experimental TBSV host range covers over 120 plant species in more than 20 different families including *N. benthamiana,* the widely used model plant in virology, RNA interference (RNAi) and vaccine production [3,4], on which TBSV infection rapidly leads to total necrotic collapse. In contrast, *Arabidopsis thaliana*, the well-established model organism in plant biology, genetics and plant–microbe interactions [5] is absent from the list of known TBSV host plants [2,6].

TBSV is a non-enveloped virus of approximately 30 nm in diameter. The icosahedral capsid of TBSV whose X-ray crystallographic structure has been resolved at 2.9 Å resolution [7,8] and is made of 180 identical coat protein (CP) subunits. The TBSV genome is monopartite, consisting of a molecule of positive sense single-stranded RNA of approximately 4.7 kb, from which two subgenomic RNA (gRNA) are produced during replication. The TBSV gRNA lacks both a 5′-cap and a 3′-poly(A) tail and possesses four open reading frames (ORFs) encoding five proteins. The proteins P33 and P92 are translated from the same 5′-proximal ORF1, P92 being produced by the translational read-through of the P33 stop codon [9,10]. Both P33 and P92 proteins play a central role in viral replication that occurs in numerous 60–70 nm vesicle-like intracellular membrane invaginations named spherules that coalesce to form a virus replication organelles (VRO) [11,12,13]. In this process, the accessory replication protein P33 directs the biogenesis of spherules on membranes from peroxisomes and the endoplasmic reticulum [11,14,15], whereas P92 acts as the RNA-dependent RNA polymerase [15,16,17]. The ORF2 encodes the CP of approximately 40 kDa, which is dispensable for local and systemic movement of TBSV in many host plants including *N. benthamiana* [10,16]. The remaining two ORFs, ORF3 and ORF4 display a nested configuration and encode the well characterized P19 suppressor of silencing (VSR) [17,18,19] and the P22 movement protein, respectively [20].

Paradoxically for a plant virus, most of our knowledge of TBSV–host interactions emanates from experiments performed in the yeast *Saccharomyces cerevisiae*. This unicellular eukaryotic model organisms with simple genetics has proven to be compatible with the replication of diverse positive sense (+) RNA viruses and some DNA viruses from humans, animals and plants, thus allowing the study of those essential activities in the viral life cycle [21]. In their seminal work, Nagy and colleagues established that the simultaneous expression of the two replicase proteins P33 and P92 supported the transcription and replication of a defective-interfering RNA from TBSV in a three-plasmid system [22]. Since then a wealth of information has been generated with respect to the precise function of P33 and P92 during replication and, more importantly, towards the identification of cellular factors involved in this complex process. Thus, over 100 yeast proteins were shown to interact with P33 or P92 and approximately 400 host proteins that could negatively or positively affect TBSV replication or recombination were identified [11]. These vast number of co-opted proteins comprise cellular translation factors, heat shock proteins, DEAD-box helicases, lipid biosynthesis and transfer enzymes, cytoskeleton components and even cellular energy-metabolism enzymes that altogether are reprogrammed to provide the optimal intracellular environment for TBSV replication [11,12,23,24,25,26].

Despite the wealth of data generated in yeast, genetic validation in plants has been limited by the lack of a TBSV isolate infecting model organism *A. thaliana*. Furthermore, the absence of the RNA interference pathway in *S. cerevisiae* imposes a strict limit on the investigation of host/TBSV interactions. In fact, in plants antiviral RNA interference and its suppression by VSR proteins play a pivotal role in the establishment and progression of infection [27,28,29]. Yeast lines with artificially reconstituted RNAi have been used to probe the role of virus replication complexes (VRCs) in protecting viral RNA from degradation [30], but the interactions of TBSV with the complex and multilayered plant RNAi machinery remain to be experimentally addressed. RNAi against (+)ssRNA viruses in *A. thaliana* is initiated when dsRNA from replication intermediates or secondary structures is processed by Dicer-like proteins DCL4 and DCL2 into 21- and 22-nt-long virus-derived small interfering RNA (vsiRNA), respectively. DCL4 acts as the main dicer against most RNA viruses investigated to date, while DCL2 can act as its surrogate [31,32,33,34]. VsiRNA are then loaded into Argonaute (AGO) proteins to form the RNA-induced silencing complex (RISC), to mediate sequence-specific degradation of viral RNA. The main AGO proteins involved in defense against ssRNA viruses in plants have been shown to be AGO1 and AGO2 [32,35,36,37]. The P19 protein of TBSV, arguably the best-studied VSR, has been shown in heterologous systems to suppress RNAi by sequestering siRNA and preventing their loading into AGO proteins [34,38]. While P19 can bind siRNA duplex products of all DCL proteins in planta [39], the very high affinity for 21 nt small RNA in vivo and in vitro [18,39,40,41] suggests that TBSV specifically relies on sequestration of DCL4 products to suppress antiviral RNAi.

In this study, we describe an isolate of TBSV that efficiently infects *A. thaliana* Col-0 ecotype, resulting in rapid and consistent systemic infection of all plants tested. We provide insight into the 3D structure of peroxisome-bound vesicles harboring VRCs in infected plants. Moreover, we identify and localize host dsRNA-binding proteins associated to TBSV VRCs. Finally, by taking advantage of the genetic tools available in *Arabidopsis*, we characterize the non-canonical RNAi response to TBSV infection in planta.

## 2. Materials and Methods

### 2.1. Plant Material

All *A. thaliana* lines used were previously described: *drb2-1*, *drb4-1* and *drb2-1/drb4-1* [42]; *dcl2-1*, *dcl4-2* and *dcl2-1/dcl4-2* [43,44]; *ago2-1* [45] and *ago1-27* [45]. The 35S:*B2:GFP*/Col-0 line was also previously described [46], as was 35S:*B2:GFP*/*N. benthamiana* [47]. *A. thaliana* were grown in a neon-lit chamber at 22–18 °C, 12 h/12 h light/dark photoperiod throughout the infection experiments, while *N. benthamiana* plants were grown in a greenhouse at 22–18 °C, 16 h/8 h light/dark photoperiod. All tissues for immunoprecipitation or molecular analysis were frozen in liquid nitrogen upon harvest and stored at −80 °C.

### 2.2. Virus Infection and Transgene Transient Expression

Lyophilized tissue from *N. glutinosa, Chenopodium quinoa* and Petunia infected with TBSV BS3 [48], used as TBSV positive control in ELISA, was obtained from Bioreba, Switzerland (Bioreba, Art-No:161853). The inoculum was propagated by rub-inoculation on *N. benthamiana* followed by the harvest of systemically infected leaves, which were used to infect *A. thaliana*. This was achieved by grinding the infected tissue in liquid nitrogen and resuspending in 50 mM sodium phosphate buffer, pH 5.8. After clearing the debris by centrifugation for 2 min at 2000× *g* and transferring the supernatant to a new tube, the resulting inoculum was gently rubbed onto *Arabidopsis* leaves previously sprinkled with celite. After a few minutes, leaves were rinsed with water. The pEAQ-ΔP19 plasmids expressing *35S:tRFP*, *35S:DRB2:tRFP*, *35S:DRB4:tRFP* and *35S:tRFP:NFD2* were previously described [46]. For overexpression and TBSV infection experiments, leaves of 5–6 week-old 35S:*B2:GFP/N. benthamiana* were infiltrated with *A. tumefaciens* GV3101 carrying the pEAQΔP19 plasmid of interest, at absorbance_600nm_ (A_600_) of 0.2. Bacteria were incubated in 10 mM MES pH 5.6, 10 mM MgCl_2_, 200 μM acetosyringone for 1 h, then infiltrated with a syringe without a needle. The following day, TBSV inoculum was applied as above on the infiltrated tissues, on the adaxial side of the leaves. Molecular/microscopy analysis was carried out 3 days after infection.

### 2.3. Virus Isolation and HTS Sequence Analyses

TBSV was purified from infected *N. benthamiana* and genomic RNA extracted as described [49]. Analyses of HTS dataset was performed using Workbench 12.0 software (CLC bio Genomics, Aarhus, Denmark), as previously described [50]. Briefly, after the trimming procedure and quality check, only reads above 70 nucleotides (nts) were kept. De novo assembly was then performed using word size = 17 and contig length min = 200 as parameters. Contigs were then tested against TBSV reference sequences and against NCBI references using BlastN/BlastX (http://blast.ncbi.nlm.nih.gov/Blast.cgi, last visited 04/2020).

Multiple sequence alignments comparing TBSV-BS3 and all available full length TBSV sequences were performed using CLUSTALW [51] and maximum likelihood-based phylogenetic trees were conducted MegaX software [52]. The best ML-fitted model for each sequence alignment (nucleic or amino acid) was used. Nodes in phylogenetic trees and branch validity were evaluated by bootstrap analyses (100 replicates).

### 2.4. Peroxisome Isolation

The detailed procedure of *Arabidopsis* peroxisome isolation is described in [39,53].

### 2.5. Immunoprecipitation

Immunoprecipitation procedures were carried out as in [46,54]. Rosette leaves (0.15 g) were ground in liquid nitrogen, homogenized in a mortar with 1 mL lysis buffer (50 mM Tris–HCl, pH 8, 50 mM NaCl and 1% Triton X-100) containing a protease inhibitor (Roche, Basel, Switzerland), transferred to a tube, incubated for 15 min at 4 °C on a wheel with slow rotation. Lysate was clarified by two successive centrifugations at 12,000× *g* for 10 min at 4 °C, after which an aliquot of supernatant was set aside on ice as input. The remaining lysate was incubated with anti-GFP magnetic beads (μMACS purification system, Miltenyi Biotech, Bergisch Gladbach, Germany, catalog number #130-091-125) at 4 °C for 20 min. A sample was passed through the M column (MACS purification system, Miltenyi Biotech) and an aliquot of the flow-through was set aside on ice. The M column was washed twice with 500 μL lysis buffer and once with 100 μL of washing buffer (20 mM Tris–HCl, pH 7.5). The beads and associated immune complexes were recovered by removing the column from the magnetic stand and passing 1 mL of Tri Reagent (for subsequent RNA analysis—see dedicated section) or 200 μL hot 1X Laemmli buffer (for protein analysis—see dedicated section). 4X Laemmli buffer was added to input and flow-through fractions before protein denaturation for 5 min at 95 °C.

### 2.6. Protein Extraction and Analysis

Immunoprecipitated proteins for mass spectrometry were isolated as described above, then directly denatured for 5 min at 95 °C. Immunoprecipitated proteins from RNA IP were obtained by transferring to a new tube 400 μL of the phenolic phase following Tri-reagent/chloroform extraction (see RNA analysis section), adding 3 vol acetone, mixing by inversion and incubating at −20 °C O/N. After centrifugation (13,000 rpm, 15 min, 4 °C) the pellet was washed in 80% acetone, resuspended in 1X Laemmli and denatured for 5 min at 95 °C. Proteins were resolved by SDS-PAGE and electroblotted onto Immobilon-P PVDF membrane. This was incubated with anti-GFP polyclonal antibody and revealed with Roche LumiLight ECL kit following incubation with the secondary antibody.

### 2.7. RNA Extraction and Analysis

RNA extraction from total and immunoprecipitated fractions was performed with Tri-Reagent (Sigma, St. Louis, USA) according to the manufacturer’s instructions. For total tissue, 0.2 g of frozen tissue were ground in liquid nitrogen and homogenized in 1 mL of Tri-Reagent, while for immunoprecipitated fractions 1 mL of Tri-reagent was passed through the columns as described above. 400 μL of chloroform was added and the sample was vortexed/shaken for 2 min. After 10 min centrifugation at 13,000 rpm, 4 °C, the supernatant was transferred to new tube and 1 vol isopropanol was added (and 1.5 μL glycogen in the case of immunoprecipitated samples) and incubated 1 h on ice (O/N at −20 °C for IP). After a 15 min spin at 13,000 rpm, 4 °C (30 min for IP), the supernatant was discarded, 350 μL of 80% ethanol was added, tubes were spun for a further 5 min, the supernatant was discarded and the pellet was dried and resuspended in water. High molecular weight Northern blot was performed by electrophoresis of 5 μg of total RNA on 1% agarose, HEPES pH 7.4, 6% formaldehyde gels followed by capillary transfer in SSC 20x onto Amersham HyBond N+ nylon membrane and UV crosslinked for 1 min. Low molecular weight Northern blot was performed by electrophoresis of 10 μg total RNA on 15% polyacrylamide, 0.5x TBE, 7.5 M urea gels followed by electro-transfer onto Amersham HyBond NX nylon membrane and 1-Ethyl-3-(3-dimethylaminopropyl)carbodiimide EDC chemical crosslinking [39]. miR159, miR173, TAS1 (ta-siRNA255) and snU6 were detected through DNA oligonucleotides labeled with γ-^32^P-ATP using T4 PNK (Thermo Fischer, Waltham, USA). TBSV genomic and subgenomic RNAs were detected in the same way, with an oligonucleotide complementary to 3′-terminal nucleotides of the viral genome. TBSV-derived and IR71 siRNA were detected through PCR products labeled by Klenow reaction (Promega, Madison, USA) in the presence of α-^32^P-dCTP. The cDNA to use for this PCR was obtained by using the SuperScriptIII kit (Invitrogen, Carlsbad, USA) with a TBSV-specific primer (same oligo used for probe) on RNA from TBSV-infected plants. Northern blots hybridization was carried out O/N at 42 °C in PerfectHyb (Sigma), followed by 3 washes of 15 min in 2X SSC, 2% SDS at 50 °C. In the Northwestern blots, dsRNA was detected through recombinant Strep-Tagged FHV B2 after migration of 5 μg total RNA at 4 °C on non-denaturing HEPES, 1% agarose gel and capillary transfer, as previously described [47]. To perform RT-qPCR, first cDNA was generated using SuperScript IV (Invitrogen) with random priming, then qPCR was performed with the SYBR-Green kit (Roche) in a LightCycler 480 Real-Time PCR System (Roche). Using *UBIQUITIN 10* (UBQ10) and *ACTINE2* mRNA as a reference, relative expression levels were calculated using the ΔΔCt method. List and sequences of primers and oligonucleotides used in this study are described (Appendix A).

### 2.8. Mass Spectrometry Protein Analysis

Mass spectrometry procedures were carried out as in [46]. Proteins were digested with sequencing-grade trypsin (Promega) and analyzed by nanoLC-MS/MS on a TripleTOF 5600 mass spectrometer (Sciex, Framingham, MA, USA), as previously described [55]. Data were searched against the TAIR v.10 database with a decoy strategy (27,281 protein forward sequences). Peptides were identified with the Mascot algorithm (version 2.5, Matrix Science, London, UK) and data were further imported into the Proline v1.4 software (http://proline.profiproteomics.fr/). Proteins were validated on Mascot pretty rank equal to 1, and 1% FDR on both peptide spectrum matches (PSM score) and protein sets (Protein Set score). The total number of MS/MS fragmentation spectra was used to quantify each protein from at least three independent biological replicates. A statistical analysis based on spectral counts was performed using a homemade R package as described in [56]. The R package uses a negative binomial GLM model based on EdgeR [57] and calculates, for each identified protein, a fold-change, a *p*-value and an adjusted *p*-value corrected using the Benjamini–Hochberg method.

### 2.9. Fluorescence Microscopy

Leaf disks from *N. benthamiana* leaves were collected 3 days after rub inoculation (4 days after agro-infiltration), placed on standard microscopy slides, covered with coverslips, immersed in water by pipetting and placed in the vacuum chamber until air had been removed. Observations of leaf disks were carried out using Zeiss LSM700 and LSM780 laser scanning confocal microscopes. eGFP was excited at 488 nm, tRFP was excited at 561 nm. Image processing was carried out with ImageJ/FIJI, while figure panels were assembled with Adobe Photoshop and Affinity Photo.

### 2.10. Electron Microscopy

*N. benthamiana* and *A. thaliana* leaves were fixed overnight at 4 °C with 3% glutaraldehyde in 0.05 M phosphate buffer saline pH 7.5. Samples were then washed 3 × 10 min before post-fixation with 1.5% potassium ferrocyanide reduced, 2% osmium for 1 h at room temperature. Leaves were then treated with filtered thiocarbohydrazide (TCH) for 20 min at RT. After 3 × 10 min washes with ddH_2_O samples were stained with 2% osmium tetroxide for 30 min and with 1% uranyl acetate overnight at 4 °C. The next day, a fresh solution of en bloc Walton’s lead aspartate solution was used to stain samples for 30 min at 60 °C (0.7% of lead nitrate in L-aspartic acid, pH 5.5). Leaves were then dehydrated with an ethanol series and infiltrated with EPON812 resin. Leaves were directly mounted on an aluminum 3View pin and embedded in 100% resin for 72 h at 60 °C. Samples were trimmed with an ultra-microtome. Side edges were silver painted with EPO-TEK H20S and sputter-coated with a 15 nm gold layer.

Backscattered electron images were acquired using a SBF scanning Zeiss Sigma VP300 electron microscope working at 41 Pa, 4 kV and equipped with a Gatan 3View2.XP. Imaging parameters for *N. benthamiana* were 2000 pixels × 2000 pixels, XYZ pixel size 4.3 nm × 4.3 nm × 70 nm, pixel time 8 microsec, while for *A. thaliana* parameters were 2000 pixels × 2000 pixels, XYZ pixel size 3.2 nm × 3.2 nm × 30 nm, pixel time 8 microsec.

DM3 files from the 3View microscope were compiled into a TIFF stack using the ImageJ-Fiji software [58]. The area of interest was cropped and segmentation was performed manually using trakEM2 [59]. The outlines of the object of interest were drawn using a Wacom graphics tablet. The 3D representation of peroxisomes was performed with the ImageJ-Fiji volume viewer plugin. Three-dimensional rendering was achieved with the display volume mode and tricubic smoothing interpolation.

## 3. Results

### 3.1. Characterization of a Tomato Bushy Stunt Virus (TBSV) Isolate Infecting Arabidopsis thaliana

Our initial aim was to identify a TBSV isolate able to infect the model plant *Arabidopsis thaliana*. To do so, TBSV isolate BS3 [48] initially propagated on *Nicotiana glutinosa*, petunia and *Chenopodium quinoa* was used as a source of inoculum and mechanically inoculated on *Arabidopsis* Col-0 ecotype. 9 days post-inoculation (dpi), newly emerged uninoculated leaves were screened by Northern blotting using as a probe the highly conserved 3′ end sequence from TBSV. All three sources of inoculum displayed infectivity on Col-0 plants (Appendix A), however only TBSV isolate BS3 initially propagated on *N. glutinosa*, named TBSV-BS3Ng hereafter, led to 100% infection rate (8/8 plants) accompanied by mild symptoms on systemic leaves (Figure 1A,B) and was therefore further characterized. Firstly, the virus was fully sequenced by high-throughput sequencing (HTS). RNA from purified virions produced >2.3 × 10^6^ clean reads of an average size of 136.8 nts, out of which > 99% mapped the de novo assembled viral genome. Due to the very high sequencing depth (>67 K), the genome sequence and organization were easily determined, confirming the virus classification in the family *Tombusviridae*, genus *Tombusvirus* (Figure 1C and Appendix A). The sequence was deposited to GenBank under accession number MT856702.

Four open reading frames (ORFs) were detected from the monopartite ssRNA, 4760 nt-long genome of TBSV-BS3Ng (Figure 1C). ORF1 encoding the putative P33 protein started from an AUG at nt 158–160 and terminated by an amber termination codon at nt 1046–1048. A read-through would extend to nt 2614 and give ORF1-RT encoding the predicted RNA polymerase. ORF2 comprises 1194 nts (position 2611 to 3804), which corresponds to the coat protein (CP) coding sequence. The deduced CP amino acid sequence showed a high identity (≥96.1%) with those from TBSV-Statice [48] and TBSV-nipplefruit [60], but fell well below the species demarcation (set at 85% identity) for TBSV-Pepper (<71.2%) and TBSV-Cherry (72.5%, Figure 1D). Due to a differential host range and identity sequence in the CP according to ICTV *Tombusviridae* guidelines (https://talk.ictvonline.org/ictv-reports/ictv_9th_report/positive-sense-rna-viruses-2011/w/posrna_viruses/277/tombusviridae, last visited 04/2020), TBSV-BS3Ng together with TBSV-nipplefruit and TBSV-statice, could be considered different species than TBSV-pepper and TBSV-cherry. ORF3 and ORF4 encoding the putative movement protein and suppressor of silencing (VSR) display the typical *Tombusvirus* nested configuration, at position 3840–4409 and 3872–4390, respectively (Figure 1C).

### 3.2. 3D electron Microscopy of TBSV Replication Vesicles on Peroxisomes from Infected Plants

All positive-strand (+) ssRNA viruses replicate their genomes by co-opting intracellular membranes and subverting host components, thereby enabling the assembly of the replication organelles. The structure, composition and formation of replication organelles vary greatly between different viruses [61,62,63]. The TBSV replication organelle has been extensively studied in various host plants and *Saccharomyces cerevisiae* [15,61,64]. Replication per se takes place inside numerous separate membrane invaginations called spherules that derive from peroxisomes and the endoplasmic reticulum [12,15,65]. To test if TBSV-BS3Ng replication is also associated to the production of spherules, we first infected *N. benthamiana* constitutively expressing the dsRNA-binding sensor protein B2:GFP [47]. Confocal and electron microscopy analyses of these plants confirmed that TBSV-BS3Ng replicates in association with severely modified peroxisomes containing numerous spherules (Appendix A and [46]). Confocal and electron microscopy analyses of constitutively-expressing B2:GFP *A. thaliana* Col-0 plants infected with TBSV-BS3Ng (Figure 2) revealed similar cytopathic structures than those observed in *N. benthamiana* (Appendix A). At low magnification, B2:GFP was redistributed from a nucleo-cytoplasmic localization (Figure 2A) to numerous fluorescent aggregates (Figure 2B) upon infection with TBSV-BS3Ng. Confocal analysis of these aggregates at higher magnification revealed numerous clustered ring-like structures very similar to those observed in *N. benthamiana* corresponding to peroxisomes (compare Figure 2C,D to Appendix A). Arrays of individual serial block face SEM (SBEM) images taken from 35S B2:GFP *A. thaliana* plants infected with TBSV-BS3Ng at 9 dpi revealed altered peroxisomes containing numerous spherules (Figure 2E,F). Three-dimensional reconstruction of a cytopathic peroxisome revealed that all spherules were embedded inside the lumen of the organelle (Figure 2G) similarly to *N. benthamiana* (Appendix A). Altogether it was concluded that TBSV-BS3Ng was able to infect *A. thaliana* in a systemic manner and its replication occurred on peroxisome-associated spherules. Our results also indicate that B2:GFP had access to the dsRNA present within spherules.

### 3.3. Immunoprecipitation of TBSV Replication Complexes

After having established that the TBSV-BS3Ng isolate forms replication vesicles on the membranes of peroxisomes in *Arabidopsis*, we sought to isolate and characterize the VRC proteome directly from purified peroxisomes. To do so, and taking advantage of our experience with the peroxisome isolation procedure [34,39], we first isolated peroxisomes from non-infected and TBSV-infected *dcl2-1/dcl4-2* mutant plants. Mass spectrometry (MS) analysis of peroxisomes isolated from healthy and TBSV-infected plants revealed numerous hits corresponding to peroxisomal proteins as expected (Appendix A). However, although peptides from the TBSV P41 and the P19 VSR were detected in the peroxisome fraction from infected plants only, the infected peroxisome extracts contained no peptides matching the TBSV replicase, the hallmark component of VRCs. The absence of replicase peptides led us to conclude that this experiment did not achieve sufficient isolation of VRCs. It is likely that the cytopathic peroxisomes (Figure 2) display different biophysical properties that prevent them from being isolated by density gradient fractionation using conventional procedures [34,39]. In addition, the large number of host proteins present in peroxisomal fractions (Appendix A and [34]) may interfere with the detection of possibly small quantities of viral proteins by mass spectrometry.

Given these results, and considering B2:GFP localizes to TBSV replication complexes in *N. benthamiana* (Appendix A and [46]) and *A. thaliana* (Figure 2), we decided to isolate VRCs using the technique we have recently developed for *Tobacco rattle virus* (TRV) [46,54]. This procedure enables the identification of TRV dsRNA-associated proteome using B2:GFP as bait for immunoprecipitation (IP). Molecular analysis of the immunoprecipitated fractions revealed that TBSV RNA (Figure 3A) and dsRNA (Figure 3B) were co-immunoprecipitated with B2:GFP, but not with GFP. Interestingly, contrarily to TBSV-infected plant expressing GFP only in which vsiRNA were abundantly produced, B2:GFP-expressing plants accumulated little if any detectable levels of TBSV-derived siRNA (Figure 3C). Finally, the Western blot analysis of IP fractions using GFP antibodies revealed that GFP and B2:GFP were specifically immunocaptured as expected (Figure 3D).

We then performed anti-GFP IP on B2:GFP/TBSV-infected leaves in triplicate, and analyzed the immunoprecipitated proteins by mass spectrometry. Nine controls from three genotypes/conditions, comprising of anti-GFP IPs performed in GFP/TBSV-infected leaves and IPs performed in non-infected B2-GFP or GFP leaves were included in the analysis. The comparison of B2:GFP IPs performed in TBSV-infected leaves to the 9 controls revealed a total of 11 proteins significantly enriched in the IPs with an adjusted *p*-value < 0.05 (Appendix A) as shown in the volcano plot representation (Figure 3E). As expected, these include the TBSV proteins P19, P92 and P41 (Figure 3E, blue spots) as well as eight host proteins: AT1G24450 (NFD2), AT3G55410, AT3G62800 DRB4), AT5G55070, AT1G52400, AT4G26910, AT1G30120 and AT2G28380 (DRB2). Remarkably, among these significantly-enriched proteins, NFD2 (Nuclear Fusion Deficient 2), DRB2 (Double-stranded RNA Binding 2) and DRB4 (Double-stranded RNA Binding 4; Figure 3E, red spots) belong also to the TRV dsRNA-associated proteome [46].

### 3.4. Host Double-Stranded RNA-Binding Proteins Associate to TBSV Replication Complexes

Considering NFD2, DRB2 and DRB4 are associated to TRV replication complexes [46] and were also found to be significantly enriched upon dsRNA IP from TBSV-infected plants (Figure 3E), we further investigated their association with TBSV replication complexes by confocal microscopy. To do so, we transiently expressed tRFP-tagged candidates in 35S:*B2:GFP*/*N. benthamiana* leaves and subsequently infected them with TBSV, as previously described [46]. In accordance with our previous report on TRV infection [46], tRFP localization was unaffected upon TBSV infection, showing a constant nuclear-cytoplasmic distribution, while B2:GFP surrounded clustered peroxisomes (Figure 4A, Appendix A). As expected, DRB2:tRFP colocalized with B2:GFP at the periphery of peroxisomes upon infection with TBSV (Appendix A and [46]). Similarly, striking relocalization of DRB4:tRFP from the nucleus in healthy cells (Appendix A and [46]) to B2-labeled peroxisome clusters was observed upon TBSV infection (Figure 4B). This relocalization of DRB4 from the nucleus to VRCs is in line with published data [46,66]. Observation at a higher magnification of the “bunch of grape”-like structures corresponding to clustered peroxisomes revealed two types or localization for B2:GFP, bright peripheral rings as well as dim intraperoxisomal punctate structures (Figure 4B, bottom panels). From our electron microscopy data it is likely that the peripheral B2 labeling corresponds to spherules connected to the outer peroxisomal membrane while internal labeling likely points to intraperoxisomal spherules (Figure 2F,G and Appendix A). Remarkably, DRB4:RFP colocalized only with the peripheral spherules (that appeared yellow upon merging of the channels) and not with the intraperoxisomal spherules (arrowheads, Figure 4B). This suggests that B2:GFP and DRB4:tRFP differ in their capacity to access spherule-associated dsRNA within peroxisomes. However, the significance of this difference remains to be determined.

Finally, as with DRB2 and DRB4, TBSV infection also strongly influenced the intracellular localization of NFD2, a putative RNAse III-Like protein also known as RTL4 [67], which concentrated in the matrix surrounding the grape-like peroxisome clusters (Figure 4C and Appendix A). In this case, no colocalization between NFD2 and B2 was observed, suggesting that NFD2 is associated with the larger viral factory but not with the spherule-restricted replication complexes per se in which genomic viral dsRNA accumulate.

Next, we wondered whether transient overexpression of tRFP-tagged DRB2, DRB4 and NFD2 in 35S:*B2:GFP*/*N. benthamiana* has any effect on TBSV accumulation. Northern blot analysis performed on RNA extracted from two independent pools of agroinoculated and infected tissues showed a drastic reduction in TBSV RNA accumulation in tissues expressing DRB2:tRFP (Figure 5A). This is well in agreement with analogous results we obtained in wild-type *N. benthamiana* [46]. In contrast, while no clear difference was observed upon DRB4:tRFP and tRFP overexpression, a slight increase in TBSV accumulation was detected upon tRFP:NFD2 overexpression (Figure 5A). We then wondered whether knock-out of these proteins could affect TBSV accumulation, viral dsRNA patterns or TBSV-derived siRNA accumulation. Unfortunately, since NFD2 homozygous mutation leads to sterility [68], mutants were not available. We therefore restricted our analysis to *drb2-1*, *drb4-1* and *drb2-1/drb4-1* mutants [42]. Northern analysis of total RNA from systemically infected leaves at 10 dpi revealed a moderate increase in TBSV RNA accumulation in all *drb2-1*, *drb4-1* and *drb2-1/drb4-1* mutants compared to wild-type plants (Figure 5B). Northwestern blot revealed no evident changes in the accumulation or patterns of dsRNA (Figure 5C). Finally, a moderate decrease in TBSV-derived siRNA was observed in lines defective in DRB4 function (*drb4*, Figure 5D), consistently with the known role of DRB4 as a player in antiviral siRNA biogenesis and a cofactor of DCL4 [42,66,69,70].

### 3.5. DCL2 Does not Effectively Process TBSV dsRNA

The bulk of genetic data available on TBSV infection was generated on surrogate host *S. cerevisiae*, which does not possess the RNA interference molecular machinery. Although the RNAi machinery has been artificially reconstituted in yeast [71] and used to investigate anti-TBSV RNAi [30], genetic data are missing in plants. We therefore sought to take advantage of the genetic tools available in *A. thaliana*, along with the body of data generated on antiviral RNAi in this species, to characterize the RNAi response to TBSV in planta. Considering that DCL2 and DCL4 proteins are among the principal actors in host antiviral defense [31], we infected Col-0, *dcl2-1*, *dcl4-2* and *dcl2/4* with TBSV and analyzed total RNA from systemic leaves (Figure 6). Northern blot analysis of RNA from independent pools of plants revealed a moderate increase of TBSV genomic RNA accumulation in the *dcl4* and *dcl2/4* lines compared to wild-type plants (Figure 6A). As upon overexpression of tRFP:NFD2 (Figure 5A), whether this observed increase in TBSV accumulation is statistically significant remains to be determined. However, the viral increase in the absence of both dicers is consistent with observations on other viruses expressing strong VSR proteins [31,32,33,34,72]. The fact that the strongest effects on TBSV genomic RNA accumulation was observed in *dcl4* and *dcl2/4* suggests that DCL2 may not participate significantly in RNAi against TBSV. Moreover, Northwestern blot analysis revealed that in the absence of DCL4 a distinct dsRNA species over-accumulated (arrow, Figure 6B). The relatively small size of this dsRNA suggested that it could be derived from one of the TBSV subgenomic RNAs produced from the 3′ portion of the virus. Since this dsRNA species was barely detected in the presence of DCL4, we reasoned that it could be a prime target for siRNA generation by DCL4. We therefore analyzed siRNA derived from subgenomic RNAs (@P19) and the genomic RNA only (mid, Figure 6C) and found that siRNA from both regions could be readily detected, suggesting that this dsRNA species is not the only substrate of DCL4. Sequencing experiments coupled with isolation and sequencing of long dsRNA are required to better elucidate the nature and processing of this dsRNA species.

Surprisingly, but in agreement with the results described above, virtually no antiviral siRNA, in particular the DCL2-derived 22 nt siRNA species could be detected upon DCL4 knock-out (Figure 6C). This indicates that DCL2 does not process TBSV dsRNA, in contrast with the hierarchical action of DCL4 and DCL2 described for other viruses [31,32,33,34].

In an attempt to confirm these findings and gain further insight into this atypical RNAi response, we repeated the TBSV infections in duplicate and included mutants for the two key Argonaute proteins involved in defense against RNA viruses, AGO1 and AGO2 [73]. Northern and Northwestern blot RNA analysis revealed again enhanced TBSV replication (Figure 7A) and the apparition of novel dsRNA species (Figure 7B, arrow) in *dcl4* and *dcl2/4* mutants. An increase in TBSV systemic accumulation could also be observed in the *ago1-27* and *ago2-1* mutants (Figure 7A), suggesting that both AGO1 and AGO2 participate in RNAi against TBSV.

Analysis of TBSV-derived siRNA also confirmed the previous results (Figure 6C), with the near complete depletion in siRNA species in plants defective in DCL4 (Figure 7C, upper panel). The highly ineffective dicing by DCL2 upon DCL4 knock-out suggests that DCL2 could be inactive during TBSV infection. To test this hypothesis, RT-qPCR quantification of *mDCL2* gene expression was performed in different *dcl* genetic backgrounds upon infection. As expected, the wild-type *DCL2* mRNA was expressed in all lines except in *dcl2* and *dcl2/4* mutants (Figure 7D). Since an antibody to efficiently detect DCL2 is not available, the accumulation of the protein in infected tissues remains to be determined. To partially bypass this issue, we tested DCL2 functionality by assessing the accumulation of 22 nt siRNA from IR71, an endogenous substrate of DCL2 [74,75]. In line with DCL2 being expressed and active upon TBVS-BS3Ng infection, IR71-derived 22 nt siRNA accumulated in healthy and infected Col-0, as well as in infected *dcl4* mutants, but not in *dcl2* and *dcl2/4* mutants (Figure 7C). In contrast, a dramatic increase in IR71 processing by DCL3 leading to the overaccumulation of 24 nt siRNA was observed with all genotypes upon TBSV infection (compare samples from infected plants to healthy Col-0 plants). Interestingly, this phenotype strongly resembles that of *drb4* mutants [75] and may result from the virus-induced relocalization of DRB4 to TBSV replication complexes during infection [46,66]. Altogether it is concluded that DCL2 is expressed and functional in TBSV-BS3Ng-infected *Arabidopsis* but is ineffective as a surrogate to DCL4 to trigger an antiviral response against TBSV-BS3Ng.

Further analysis of endogenous small RNA showed that TBSV had no effect on ta-siRNA processing by DCL4, and that smaller additional species of miR159 and miR173 accumulated in infected tissues (Figure 7C). These “trimmed” miRNA molecules are highly reminiscent of those observed in transgenic plants expressing the CymRSV and TBSV P19 proteins in *N. benthamiana* and *A. thaliana*, respectively [39,76]. Finally, accumulation of TBSV-derived siRNA was reduced in the hypomorphic *ago1-27* mutant compared to wild-type and *ago2-1* mutant (Figure 7E). This could be due to changes in dicing, small RNA stability and/or AGO1 loading, for example. Further experiments on a battery of RNAi mutant combinations will hopefully provide further insight into the molecular events taking place during the atypical RNAi response to TBSV in planta.

## 4. Discussion

In this study we described a new isolate of TBSV that infects *A. thaliana*. This should prove to be an extremely valuable asset in the genetic investigation of host–*Tombusvirus* interactions, which for the moment have been investigated essentially using yeast and *N. benthamiana*. The localization of the VRCs on *Arabidopsis* peroxisomes suggests that their formation and function may follow similar pathways as those dissected using yeast, reinforcing the potential of this TBSV-BS3Ng isolate as an investigative tool. One aspect of TBSV–host interaction that cannot be thoroughly studied in yeast is antiviral RNAi, which is a central player in plant antiviral defenses. Our experiments have shown that the *Arabidopsis* RNAi reaction to TBSV is unique when compared to other RNA viruses. While DCL2 has been shown to act as a surrogate to DCL4 against TRV, TCV, TuMV and CMV [31,32,33], or in parallel to DCL4 against PCV [34], during TBSV infection DCL2 is unable to efficiently generate vsiRNA. Since DCL2 is present in TBSV-infected tissues and able to process endogenous targets, we proposed that either (i) DCL2 is unable to access TBSV dsRNA and/or to process it, or (ii) TBSV inhibits a putative antiviral pool of DCL2 that is distinct from that processing IR71. Given that DCL2 is arguably the least known among the *Arabidopsis* dicer proteins, there is little data to help explain these observations. However, future investigation into this phenomenon may shed light on the activity of DCL2 as an antiviral effector.

The possibility of TBSV-BS3Ng infection inhibiting the antiviral pool of DCL2 is compelling. It would imply that TBSV has evolved a layered RNAi suppression mechanism, where the primary DCL4-dependent RNAi reaction is neutralized through P19-mediated sequestration of 21 nt vsiRNA, while the “back-up” DCL2-dependent RNAi is also somehow neutralized upstream of dicing likely by a TBSV-encoded protein that remains to be identified. Furthermore, the exclusive processing by DCL4 may explain why P19 has evolved highest binding affinity for 21 nt-long RNA [19,39,40,41]. Along with the observation that PCV, which is abundantly processed by DCL2, has evolved to efficiently sequester 22 nt siRNA through its P15 VSR [34], these findings suggest that the VSR proteins of different viruses adapt to neutralize the specific dicer products driving antiviral RNAi.

The inability of DCL2 to generate 22 nt TBSV vsiRNA is in agreement with the absence of vsiRNA in TBSV-infected 35S:*B2:GFP*/Col-0 plants. In these plants, we have previously shown that infection by TRV leads to the production of DCL2-dependent 22nt vsiRNA, as opposed to wild-type plants where vsiRNA are produced mainly by DCL4, suggesting an inhibitory activity of B2:GFP on DCL4 [46]. Upon TBSV infection, DCL4 inhibition by B2 and DCL2 inability to generate vsiRNA likely leads to the observed absence of 21-22 nt-long vsiRNA. The identification of DRB2 and DRB4 associated to VRCs in 35S:*B2:GFP*/Col-0 plants suggests that their binding of dsRNA is vsiRNA-independent.

Little is known about the viral dsRNA substrates of DCL4. Many works have investigated the distribution of vsiRNA along viral genomes in many species (reviewed in [77]), but these may be heavily biased toward those siRNA species that are highly stable. As we have shown, the homeostasis of viral long dsRNA in relation to dicer proteins and other RNAi components can be easily assessed by Northwestern blotting. Our results indicate that during infection DCL4 is responsible for the depletion of a dsRNA of reduced length, presumably through dicing into vsiRNA of a double-stranded subgenomic RNA. Alternatively, this could be a product of host-encoded RDR proteins that is stabilized by the absence of DCL4. However, since Northwestern blotting does not allow one to distinguish RNA sequence, it cannot be ruled out that the dsRNA species observed is actually host-encoded. This could explain why no subgenomic RNA was seen by Northern blot to over-accumulate in *dcl4* mutants compared to wild-type plants. This discrepancy between Northern and Northwestern blots could also be explained if DCL4 were depleting dsRNA while not destroying the corresponding ssRNA, for example through exclusive helicase activity. Further work will hopefully unravel the molecular mechanisms behind these observations.

Our results obtained with B2:GFP IPs followed by mass spectrometry further validate the potential of this experimental approach to characterize VRC proteomes in planta. Our microscopy experiments on DRB2, DRB4 and NFD2 are highly consistent with what we observed during TRV infection [46], and provide compelling and detailed snapshots of the TBSV VRCs on cytopathic peroxisomes. Furthermore, the results obtained with DRB2 and DRB4 are in agreement with previous reports [66,77]. We have previously described the broad-ranged antiviral activity of *Arabidopsis* DRB2 [46]. The results we obtained through overexpression of DRB2 are in contrast with similar experiments on *N. benthamiana* DRB2B, which when overexpressed increases the accumulation of PVX [78]. However, the authors of this study also show that silencing of DRB2B causes an increase in systemic infectivity of PVX, uncovering an antiviral activity of this protein that is well in agreement with our observation on *Arabidopsis* DRB2. Our observations that DRB4 was not strictly necessary for DCL4-dependent production of TBSV-derived siRNA is in agreement with published studies on TuMV and TCV [42,70]. However, DRB4 has been shown to be mandatory for DCL4 activity during infection by TSWV and TYMV [42,66], suggesting that DRB4-DCL4 interplay is highly dependent on virus species. Interestingly, the moderate increase in TBSV accumulation observed in *drb4* mutants is reminiscent of that observed in *dcl4* mutants, suggesting that either (i) the vsiRNA produced by DCL4 in the absence of DRB4 are not functional in mediating antiviral RNAi or (ii) DRB4 plays an antiviral role independently of DCL4, or both.

Our identification of NFD2 in association with dsRNA during infection by both TRV and TBSV, and its localization in close proximity to VRCs, suggests that this protein may play important roles in the viral life cycle as a putative RNAseIII. The observation that NFD2 overexpression leads to increased TBSV accumulation is in agreement with the report that overexpression of RTL1, a related RNAse III-like protein, leads to increased accumulation of TCV, TVCV and TYMV in *Arabidopsis* [79], most likely by depriving Dicer enzymes of substrate dsRNA.

The fact that B2:GFP blocks dicer activity on TBSV dsRNA, as commented above, means that B2:GFP IPs are limited in their use to investigate the association of RNAi factors to TBSV VRCs. This problem could be circumvented by developing the use of a recombinant B2 to isolate VRCs from wild-type infected plants ex vivo. Furthermore, the gradient fractionation of cytopathic peroxisomes, which was not further pursued in this study, could be optimized to allow molecular analysis of the VRC-containing vesicles present on their surface.

## Figures and Tables

**Figure 1 viruses-12-01121-f001:**
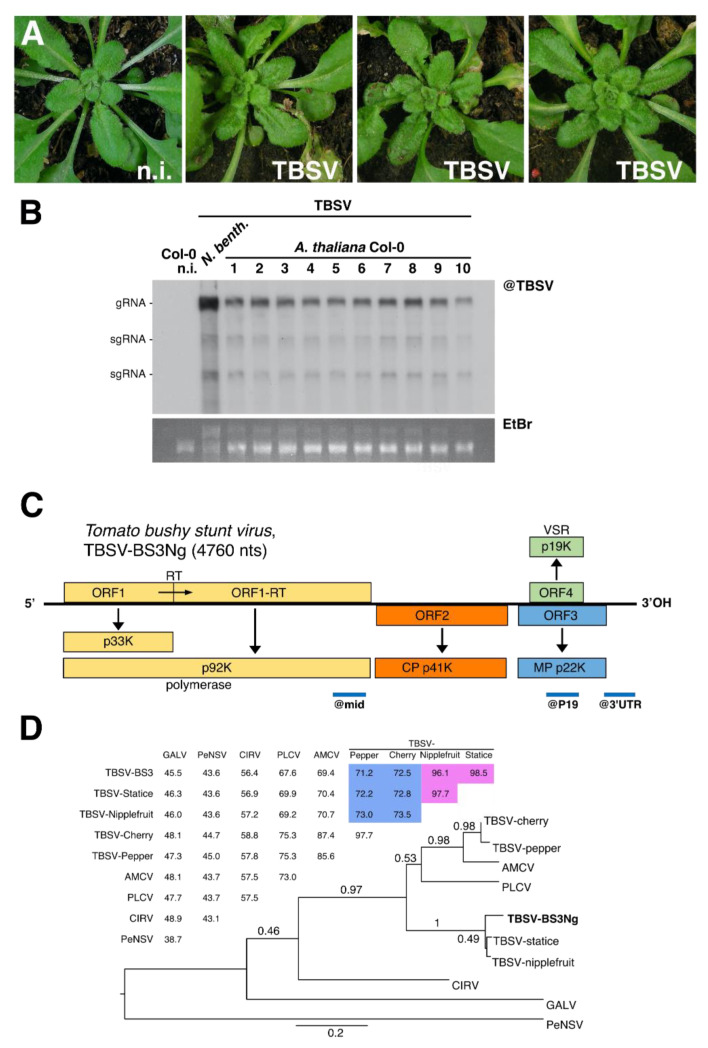
Characterization of a *Tomato bushy stunt virus* (TBSV) isolate infecting *Arabidopsis thaliana*. (**A**) *A. thaliana* Col-0 plants systemically infected by TBSV, 9 days post-infection (dpi). A non-infected (n.i.) plant is shown on the far left. (**B**) Northern blot analysis of RNA from systemically infected leaves from individual Col-0 plants 9 days after infection with TBSV. Out of the inoculated plants 10/10 resulted to be systemically infected. On the left, RNA from non-infected Col-0 and TBSV-infected *N. benthamiana* were used as negative and positive controls, respectively. Ethidium bromide staining (EtBr, bottom) was used as a loading control. (**C**) Schematic representation of the genomic organization of TBSV. Boxes represent predicted ORFs encoding the p33K and phylogenetically conserved polymerase (ORF1), the capsid protein (CP, ORF2), the movement protein (MP, ORF3) and the silencing suppressor (VSR, ORF4). RT: translational read-through of the termination codon. The blue lines below indicate the genomic segments detected by the probes used to detect small RNA in this study, referred to as @mid, @P19 and @3′UTR. (**D**) Maximum likelihood tree inferred from 10 complete CP amino-acid sequences of different *Tombusvirus* species. Table shows the CP amino acid identity percentage. The intra-TBSV species percentages are highlighted in pink and those falling outside the species demarcation are highlighted in blue. The scale bar corresponds to the number of substitutions per site.

**Figure 2 viruses-12-01121-f002:**
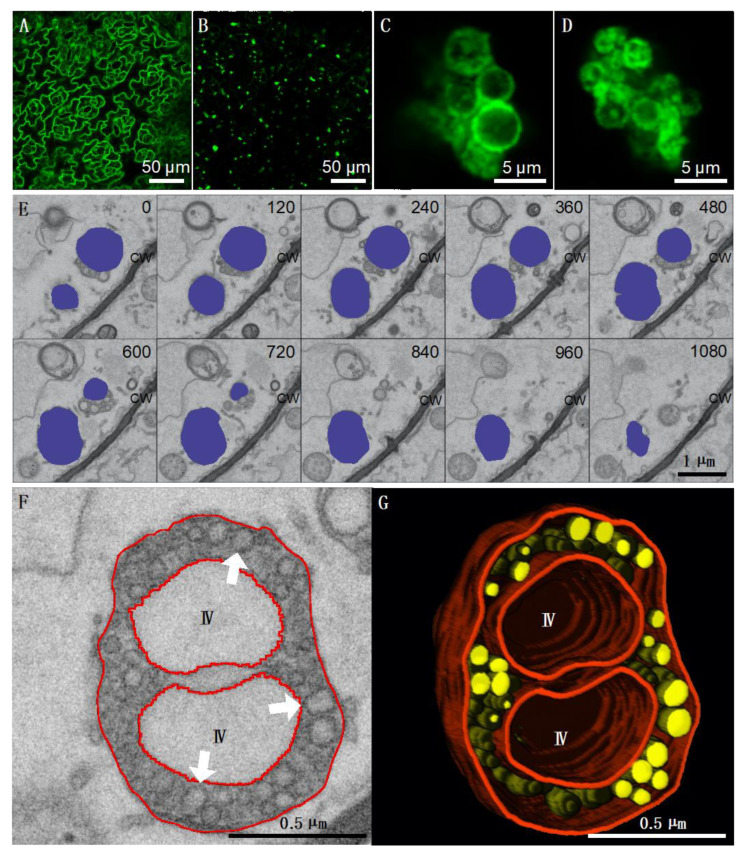
TBSV replicates in association with peroxisomes in *Arabidopsis thaliana*. Confocal microscopy imaging of leaf epidermal cells from healthy (**A**) and TBSV-infected (**B**–**D**) 35S:B2:GFP *A. thaliana* Col-0 plants at 9 dpi. At low magnification (**A**,**B**), B2:GFP adopts a nucleo-cytoplasmic localization in healthy cells (**A**), whereas in TBSV-infected tissues, B2:GFP is relocalized to numerous cytoplasmic aggregates (**B**). At high magnification (**C**,**D**), B2:GFP is essentially found in association with clustered ring-like structures. (**E**) Array of individual serial block face SEM (SBEM) images taken from 35S:B2:GFP *A. thaliana* plants infected with TBSV at 9 dpi. Image planes were recorded at 30 nm intervals, and the entire acquisition was recorded using 150 image planes. Ten sample slices taken at 120 nm intervals are shown in (**E**). Altered peroxisomes containing numerous spherules are shaded in blue. (**F**) Detail of an altered peroxisome in which numerous spherules are visible (white arrows). (**G**) Partial 3D reconstruction of the same peroxisome in which spherules are represented in yellow and membranes in red. IV: Intraperoxisomal vacuoles. CW: Cell wall.

**Figure 3 viruses-12-01121-f003:**
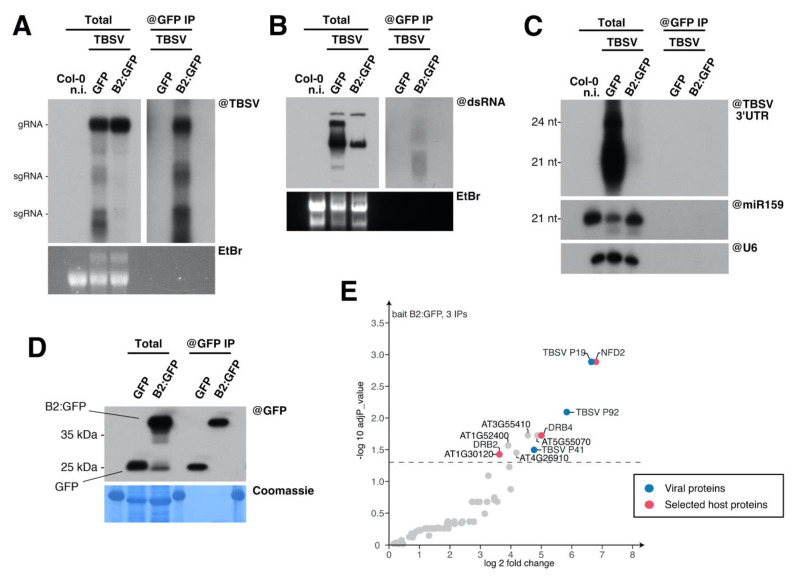
Immunocapture of TBSV dsRNA and associated proteins. (**A**) Northern blot analysis of high molecular weight RNA to detect TBSV genomic RNA. RNA was extracted from total (left) and anti-GFP immunoprecipitated (right) fractions of 35S:*GFP* and 35S:*B2:GFP*/Col-0 *A. thaliana* systemically infected with TBSV. A non-infected Col-0 control was included among the total fractions, to the far left. (**B**) Northwestern blot analysis of the RNA samples described in (**A**), size-separated on a non-denaturing agarose gel, to detect double-stranded RNA through recombinant B2-Strep. In (**A**) and (**B**) ethidium bromide staining of the gels was used as loading control. (**C**) Northern blot analysis of low molecular weight RNA from the samples described in (**A**), to detect TBSV-derived siRNA. As loading control, the same membrane was separately hybridized to probes specific to miR159 and snU6. (**D**) Western blot analysis of proteins from the samples described in (**A**) to detect GFP. Here, the non-infected Col-0 present in (**A**–**C**) was not included. (**E**) Volcano plot representation shows the enrichment of proteins from TBSV-infected plants that co-purified with B2:GFP. *Y*- and *X*-axis display adjusted *p*-values and fold changes, respectively. The dashed line indicates the threshold above which proteins are significantly enriched (adjP  <  0.05). The source data are available in Appendix A.

**Figure 4 viruses-12-01121-f004:**
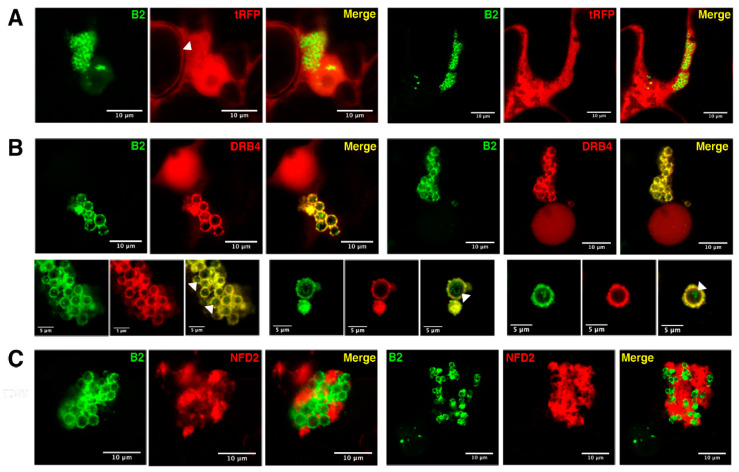
Localization of *A. thaliana* DRB4 to TBSV replication complexes. Laser confocal microscopy on 35S:*B2:GFP/N. benthamiana* leaves inoculated with *A. tumefaciens* carrying the construct of interest then infected with TBSV. For each acquisition we show the GFP channel on the left (B2), the tRFP channel in the middle (construct of interest) and a merge of the two on the right. (**A**) Observation of tissues expressing 35S:*tRFP*. (**B**) Observation of tissues expressing 35S:*DRB4:tRFP*. The lower row of three acquisitions is shown to highlight the intraperoxisomal B2 foci (white arrowheads). For the sake of simplicity, the labeling was omitted on these but is identical to that in the upper row. (**C**) Observation of tissues expressing 35S:*tRFP:NFD2*. All acquisitions were performed with a 63× objective. Scale bars indicate 5–10 μm, as indicated. Lower magnification acquisitions from these tissues, along with the respective non-infected controls, can be found in Appendix A.

**Figure 5 viruses-12-01121-f005:**
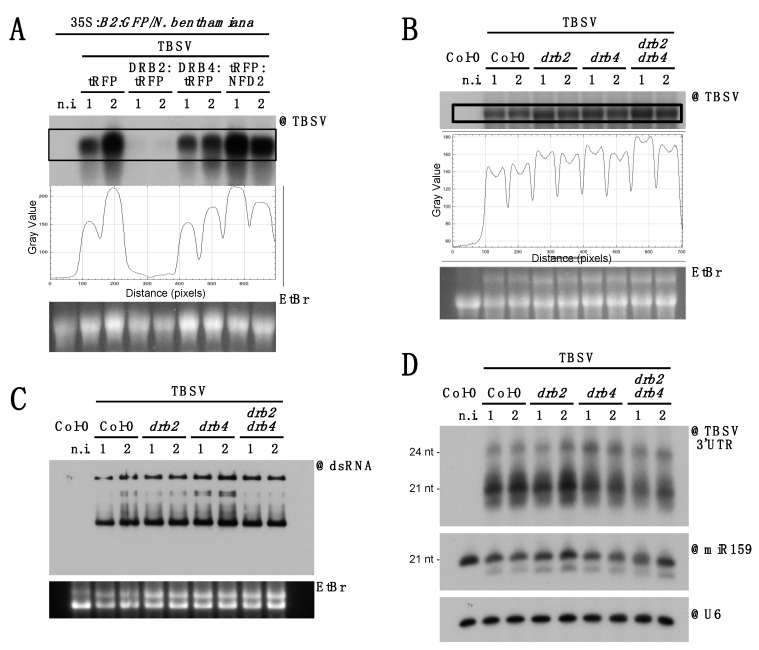
Involvement of DRB2, DRB4 and NFD2 in TBSV infection. (**A**) Northern blot analysis of RNA from 35S:*B2:GFP*/*N. benthamiana* leaf disks agroinoculated with the constructs indicated and subsequently infected with TBSV. Each sample is a pool of 50–60 leaf disks taken from several infected leaves on 2–3 plants. (**B**) Northern blot analysis of high molecular weight RNA to detect TBSV at 9 dpi in systemically infected rosette leaves of *A. thaliana* mutant lines. Quantification of signal intensity is provided. Each sample is a pool of 4–5 plants. (**C**) Northwestern blot to detect dsRNA in the samples described in (**B**). In (**A**–**C**) EtBr gel staining was used as a loading control. (**D**) Northern blot analysis of low molecular weight RNA to detect TBSV-derived siRNA. As loading control, the same membrane was separately hybridized to miR159- and snU6-specific probes. In all the blots a non-infected control (n.i.) was included on the far left. Except for the non-infected control, two samples were analyzed per genotype/transgene (1 and 2 in the labels).

**Figure 6 viruses-12-01121-f006:**
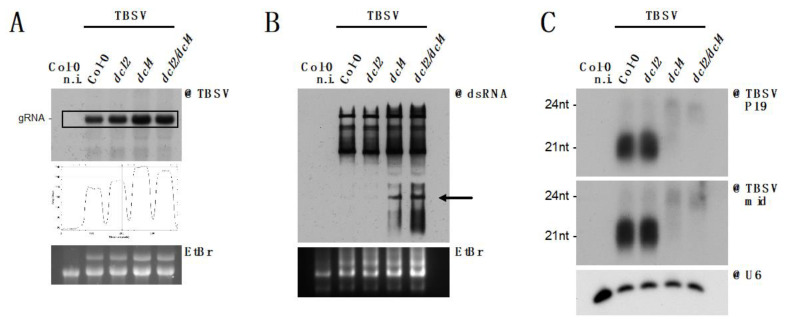
Processing of TBSV dsRNA by DCL4, but not DCL2, in *A. thaliana*. (**A**) Northern blot analysis of high molecular weight RNA to detect TBSV in systemically infected rosette leaves, 9 dpi, of *A. thaliana* mutant lines. Quantification of signal intensity is provided. Each sample is a pool of 4–5 plants. (**B**) Northwestern blot to detect dsRNA in the samples described in (**A**). The black arrow indicates the dsRNA band observed upon genetic knock-out of *DCL4*. In (**A**,**B**) EtBr gel staining was used as loading control. (**C**) Northern blot analysis of low molecular weight RNA to detect TBSV-derived siRNA, using two separate probes. TBSV-mid detects siRNA derived from the 3′ end of the P92 ORF, while TBSV-P19 detects siRNA derived from the P19 ORF. As a loading control, the membrane was separately hybridized to a snU6-specific probe. In all the blots a non-infected control (n.i.) was included on the far left.

**Figure 7 viruses-12-01121-f007:**
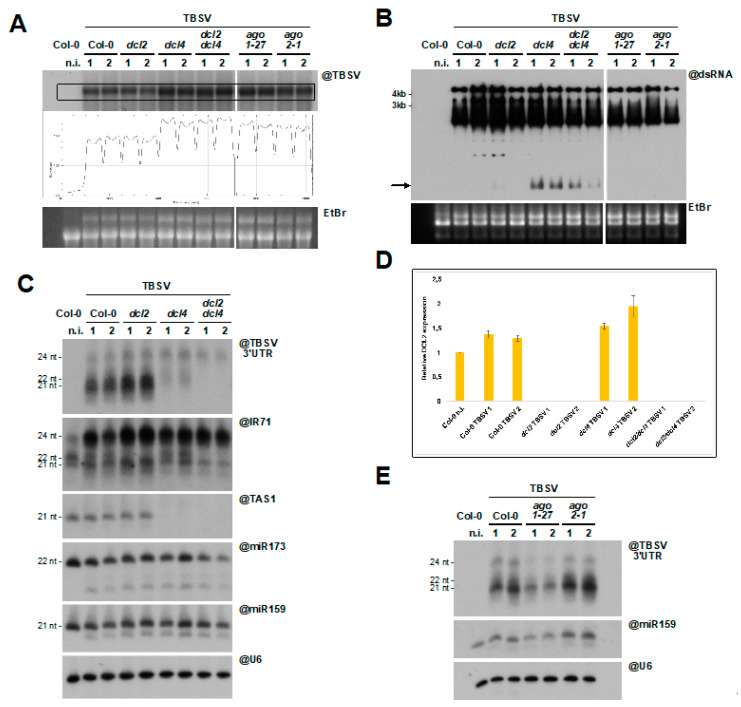
Atypical RNA silencing response to TBSV infection in *A. thaliana*. (**A**) Northern blot of high molecular weight RNA to detect TBSV in systemically infected rosette leaves, 10 dpi, of *A. thaliana* mutant lines. Quantification of signal intensity is provided. Each sample is a pool of 4–5 plants. (**B**) Northwestern blot to detect dsRNA in the samples described in (**A**). The black arrow indicates the dsRNA band observed upon genetic knock-out of *DCL4*. In (**A**,**B**) EtBr gel staining was used as a loading control. (**C**) Northern blot analysis of low molecular weight RNA to detect TBSV-derived and endogenous small RNA. The same membrane was repeatedly stripped and re-probed. (**D**) RT-qPCR of the samples analyzed in (**A**–**C**) to assess accumulation of *DCL2* mRNA. Expression levels were normalized to *UBIQUITIN 10* (UBQ10) and *ACTINE2* mRNA. Primers were designed on the opposite sides of the *dcl2-1* T-DNA insertion to rule out non-specific amplification. (**E**) Northern blot analysis of low molecular weight RNA to detect TBSV-derived siRNA in the *ago* mutants analyzed in (**A**,**B**). As in (**C**), miR159 and snU6 were used as loading controls.

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
