# Peer review of "Characterization of a DCL2-Insensitive Tomato Bushy Stunt Virus Isolate Infecting Arabidopsis thaliana"

_viruses, 2020, doi:10.3390/v12101121_

Round 1
Reviewer 1 Report
The paper describes the identification and deep characterization of a TBSV isolate able to infect arabidopsis plants. This is quite a breakthrough, as TBSV is a plant virus used as a model for numerous studies, but validation of the observations in a tractable model plant as is arabidopsis was missing. The experiments were well conducted and conclusions are solid. The amount of data generated is impressive. In general terms, it is a good manuscript that describes interesting information and clearly deserves publication. I only found a few minor aspects that deserve the attention of the authors:
- Given that the studied virus was isolated from the TBSV-BS3 sample after passaging in Nicotiana glutinosa, I would rename it to TBSV-BS3Ng, to avoid future confusions. I know that this name is a little bit long, but keeping TBSV-BS3 may be misleading, as it seems that the actual TBSV-BS3 sample is complex and only the virus in it systemically infecting N. glutinosa seems to have the features described in this paper.
- Authors need to be careful when referring to strains and isolates. In fact, "strain" is a concept, not an actual thing (i.e. a strain does not infect a plant). Please, correct this all along the text.
- There are a number of typos and writing mistakes that need to be corrected. I'm attaching an annotated pdf marking some of them.

Author Response
Point-by-point response to the comments by Reviewer 1
Reviewer comments are in italics, our responses in regular type.
The paper describes the identification and deep characterization of a TBSV isolate able to infect arabidopsis plants. This is quite a breakthrough, as TBSV is a plant virus used as a model for numerous studies, but validation of the observations in a tractable model plant as is arabidopsis was missing. The experiments were well conducted and conclusions are solid. The amount of data generated is impressive. In general terms, it is a good manuscript that describes interesting information and clearly deserves publication. I only found a few minor aspects that deserve the attention of the authors:
Given that the studied virus was isolated from the TBSV-BS3 sample after passaging in Nicotiana glutinosa, I would rename it to TBSV-BS3Ng, to avoid future confusions. I know that this name is a little bit long, but keeping TBSV-BS3 may be misleading, as it seems that the actual TBSV-BS3 sample is complex and only the virus in it systemically infecting N. glutinosa seems to have the features described in this paper.
We agree with this reviewer and changed the isolate name to BS3Ng where appropriate.
Authors need to be careful when referring to strains and isolates. In fact, "strain" is a concept, not an actual thing (i.e. a strain does not infect a plant). Please, correct this all along the text.
Indeed, we have incorrectly used the two words interchangeably. We now refer to TBSV-BS3Ng isolate throughout the manuscript.
There are a number of typos and writing mistakes that need to be corrected. I'm attaching an annotated pdf marking some of them.
We thank this reviewer for the corrections and positive comments. They have been addressed, except for “plant-Tombusvirus interactions” (line 541). This is what we mean to say in this instance.

Reviewer 2 Report
The manuscript of Incarbone et al. describes the identification and initial characterization of a TBSV isolate capable of infecting A. thaliana. The authors also provide some preliminary results on the unusual RNAi response elicited by the virus in planta. Considering that detailed molecular characterization of Tombusviruses has so far been restricted to the heterologous yeast system and the genetically intractable host N. benthamiana, the isolation of the TBSV-BS3 isolate capable of infecting A. thaliana could be of considerable interest. There are however, several major issues regarding the presented data, which have to be dealt with before the manuscript can be considered for publication.
Major points:
- The densitometric scans shown on figures 5, 6 and 7 can hardly be interpreted as a reliable measure for the accumulation of viral RNA. There are significant variabilities even between the parallel samples, sometimes even bigger than between samples from the different treatment groups, where differences are claimed to be exist. For example, on figure 6A in the dcl4 and dcl2/4 samples the virus RNA is claimed to the higher than in the other genotypes. Considering however, the increased loading (which is obvious by looking at the EtBr image) this is unfounded. These issues have to be solved somehow. It is also important to state, how many times the given experiments were repeated with comparable outcomes.
- DCL4 seems to play a dominant role in the RNAi response elicited by TBSV-BS3 as evidenced by the small RNA Northern blot shown on figure 6 C. Interestingly however, the effect of DRB4 seems to be negligible, if any based on the Northern blot shown on figure 5 D. Why is that? DRB4 is an obligate binding partner for DCL4 and has been proven to be essential for its activity. How is it possible that in the drb4 mutant plant vsiRNA production is not affected? Also on figure 5 D, vsiRNAs of different size classes could be seen, which may correspond to 21, 22 and 24 nt vsiRNAs. Are these degradation products? Some explanation for these discrepancies has to be provided. It would also be important to use the same probe for vsiRNA detection on figure 6 C and 5 D to be able to compare vsiRNA accumulation between plants of different genotypes (especially between drb4 and dcl4 mutants).
- To demonstrate the activity of DCL2, accumulation of IR71-derived siRNAs is shown on the Northern blot on figure 7 C. Indeed, in the dcl2 mutants the 22 nt siRNA species is disappearing however, the 21 nt siRNAs seem to persist in the dcl4 mutant plants. How is that possible? Are these degradation products? If so how one can reliably distinguish them from bona fide DCL4 products?
- In B2-GFP overexpressing plants vsiRNA production seems to be almost completely abolished as evidenced by the Northern blot on figure 3 C. This observation suggests that the high affinity interaction of B2-GFP with double-stranded virus RNA can completely compete out DCL complexes. This observation raises concerns regarding the validity of the B2-GFP affinity approach to purify intact VRCs. Most likely B2-GFP can interfere with not only DCL-virus RNA interactions, but titrate out other cellular factors as well. Although, the authors mention briefly this issue at the end of the Discussion section, this may represent a more serious problem than stated.
Minor points:
- Important references are missing. The association of DRB proteins with VRCs has been described by Barton et al. (Barton et al. 2017 Mol Plant Microbe Interact 30:435-443). The involvement of DRB2 in antiviral responses has also been reported earlier by Fatyol et al. (Fatyol et al. Journal of Virology vol. 94 Issue 11 e00017-20). These references have to be cited.
- On figure 3 D some labeling on the Western blot is missing.
- The labeling of some of the precipitated proteins is mixed up in lines 365-366.
- On figure 6 C the U6 loading control is missing.
Author Response
Point-by-point response to the comments by Reviewer 2
Reviewer comments are in italics, our responses in regular type.
The manuscript of Incarbone et al. describes the identification and initial characterization of a TBSV isolate capable of infecting A. thaliana. The authors also provide some preliminary results on the unusual RNAi response elicited by the virus in planta. Considering that detailed molecular characterization of Tombusviruses has so far been restricted to the heterologous yeast system and the genetically intractable host N. benthamiana, the isolation of the TBSV-BS3 isolate capable of infecting A. thaliana could be of considerable interest. There are however, several major issues regarding the presented data, which have to be dealt with before the manuscript can be considered for publication.
Major points:
The densitometric scans shown on figures 5, 6 and 7 can hardly be interpreted as a reliable measure for the accumulation of viral RNA. There are significant variabilities even between the parallel samples, sometimes even bigger than between samples from the different treatment groups, where differences are claimed to be exist. For example, on figure 6A in the dcl4 and dcl2/4 samples the virus RNA is claimed to the higher than in the other genotypes. Considering however, the increased loading (which is obvious by looking at the EtBr image) this is unfounded. These issues have to be solved somehow. It is also important to state, how many times the given experiments were repeated with comparable outcomes.
We agree with this reviewer that the claimed differences in virus accumulation are debatable, although in the case of dcl mutants they are consistent among different experiments and replicates (Figures 6 and 7). In consequence we have tamed our conclusions and modified the text accordingly, in particular by stating “Whether this increase in TBSV accumulation is statistically significant remains to be determined" (Lines 461-463). However, we would like to point out that some issues raised here, such as the differences between replicates, are due to the fact that we are including 2 “twin” replicates per sample. This is double what is normally considered valid for publication in blotting experiments, that is one sample (containing a pool of several plants) in the majority of papers. So, the variation seen here between twin replicates should not be considered a limit of this paper, but a limit of the commonly accepted format of one sample per genotype/condition, where such differences would have not emerged.
DCL4 seems to play a dominant role in the RNAi response elicited by TBSV-BS3 as evidenced by the small RNA Northern blot shown on figure 6 C. Interestingly however, the effect of DRB4 seems to be negligible, if any based on the Northern blot shown on figure 5 D. Why is that? DRB4 is an obligate binding partner for DCL4 and has been proven to be essential for its activity. How is it possible that in the drb4 mutant plant vsiRNA production is not affected?
We disagree with this reviewer concerning the essentiality of DRB4 in all activity of DCL4. Indeed, DRB4 has been shown to be essential for DCL4 activity in vitro (Fukudome et al., 2011). However in vivo, while it is necessary for DCL4 processing of miR822, it is partially dispensable for TAS1 and TAS3 processing (Montavon et al., 2017). DRB4 is necessary for DCL4 processing of TYMV (Jakubiec et al, 2012) and TSWV (Curtin et al, 2008), but partially dispensable for processing of TuMV (Curtin et al, 2008) and completely dispensable for processing of TCV (Qu et al., 2008, see supplementary information there). Therefore, we argue that the role DRB4 plays in DCL4 activity depends on conditions and parameters that remain to be elucidated. Furthermore, this result does not impact the validity of the data shown here.
Also on figure 5 D, vsiRNAs of different size classes could be seen, which may correspond to 21, 22 and 24 nt vsiRNAs. Are these degradation products? Some explanation for these discrepancies has to be provided. It would also be important to use the same probe for vsiRNA detection on figure 6 C and 5 D to be able to compare vsiRNA accumulation between plants of different genotypes (especially between drb4 and dcl4 mutants).
The two main TBSV-derived siRNA bands observed are 21nt (major) and 24nt-long (weaker). Especially the 21nt-long bands show a “tail” of signal both above and below the main band. As this reviewer suggests, this could be some kind of degradation product. While this is possible, we do not believe it to be the case. Following many similar results with different viruses using these large high-resolution PAGE gels, we believe this effect is caused by the probing method. As can be observed in figures 5 and 7, the same membrane probed with a labeled oligonucleotide, detecting one specific 21nt sRNA sequence (@159, @173 and @TAS1), yielded sharp bands. The Klenow-labeled PCR products used to detect virus-derived siRNA, covering 300-400 nucleotides of viral sequence, detect a very heterogeneous population of siRNA. Possibly, the differences in sequence and CG content could cause small variations in migration speed that over the length of this kind of gel result in a signal that is more spread. We have observed this with TBSV, TRV and PCV, and interestingly probes on different segments of the viral genome show different degrees of “smear” on the same membrane (this is why here we favored the cleaner @3’UTR over @P19 and @mid). The “smear” can also change from one gel to the other (see Figure 5D vs Figure 7C @TBSV 3’UTR). Of course, this evidence is anecdotal, our interpretation of this repeated pattern. To obtain sharp vsiRNA bands it would be necessary to perform small RNA sequencing and search for a vsiRNA species that is very abundant to detect through an oligo. This would be outside the scope of this work.
Concerning the last point, to compare the vsiRNA patterns in dcl4 and drb4 mutants with the same probe, please compare Figure 5D with Figure 7C.
To demonstrate the activity of DCL2, accumulation of IR71-derived siRNAs is shown on the Northern blot on figure 7 C. Indeed, in the dcl2 mutants the 22 nt siRNA species is disappearing however, the 21 nt siRNAs seem to persist in the dcl4 mutant plants. How is that possible? Are these degradation products? If so how one can reliably distinguish them from bona fide DCL4 products?
Since the absence of DCL4 can be observed in dcl4 and dcl2/dcl4 through the absence of both TAS1- and TBSV-derived 21nt siRNA, we conclude these IR71-derived 21nt siRNA are most likely products of DCL1. Of course, though unlikely, one cannot exclude these are products of degradation (not present in TBSV-derived siRNA) or some other unidentified cellular process. We do not see how this impacts the validity of our results.
In B2-GFP overexpressing plants vsiRNA production seems to be almost completely abolished as evidenced by the Northern blot on figure 3 C. This observation suggests that the high affinity interaction of B2-GFP with double-stranded virus RNA can completely compete out DCL complexes. This observation raises concerns regarding the validity of the B2-GFP affinity approach to purify intact VRCs. Most likely B2-GFP can interfere with not only DCL-virus RNA interactions, but titrate out other cellular factors as well. Although, the authors mention briefly this issue at the end of the Discussion section, this may represent a more serious problem than stated.
As we state in the text, we fully agree with this reviewer that the 35S:B2:GFP system used here can generate biases due to the effect of B2 VSR activity and binding to dsRNA in vivo, the most evident of which is a switch from DCL4- to DCL2-mediated RNAi (see case of TRV). However, we argue that any dsRNA-binding protein over-expressed in planta would impact the viral life cycle in one way or another (for an example, see the opposite yet dramatic effects of DRB2 homeolog overexpression in Fatyol et al, 2020 and our previous pre-print referenced in the paper – Incarbone et al. 2020). To address this issue, we are currently working on generating and characterizing lines expressing B2:GFP under weaker and/or tissue-specific promoters. The use of B2 in vitro, which we are also developing, would of course nullify the issue and represent the ideal tool. In the frame of this manuscript, it is evident that TBSV is equally able to form functional replication complexes, replicate and accumulate in the presence of B2, in both Arabidopsis and N. benthamiana. Therefore, we argue that, even though there may be some changes in the VRCs due to B2, the replication complexes remain equally functional and consequently interesting to investigate further.
Minor points:
Important references are missing. The association of DRB proteins with VRCs has been described by Barton et al. (Barton et al. 2017 Mol Plant Microbe Interact 30:435-443). The involvement of DRB2 in antiviral responses has also been reported earlier by Fatyol et al. (Fatyol et al. Journal of Virology vol. 94 Issue 11 e00017-20). These references have to be cited.
Indeed, we thank this reviewer for bringing this up. We cited the Barton et al paper several times in our previous paper and somehow forgot to cite it here. Fatyol et al 2020 was published in the advanced stages of preparation of this manuscript, and we also failed to include it. These references have been added (lines 566-567, 568-570).
On figure 3 D some labeling on the Western blot is missing.
Indeed, the “Total” and “IPed” labels. This has been corrected.
The labeling of some of the precipitated proteins is mixed up in lines 361-364.
Thank you, this has been corrected (lines 361-364). Acronyms were left only for the genes further analyzed.
On figure 6 C the U6 loading control is missing.
Indeed, we apologize. This has been added.

Reviewer 3 Report
Tomato bushy stunt virus (TBSV) is a positive-strand RNA virus for which abundant knowledge about RNA replication mechanisms has been obtained by using Saccharomyces cerevisiae as a model host. Although some studies on TBSV-plant interactions have been performed using a plant Nicotiana benthamiana, there was no report about TBSV infection in Arabidopsis thaliana, the most well-established model plant for genetic analyses. In this study, the authors found that the BS3 strain of TBSV is highly infectious on A. thaliana, and using this pathosystem, they show that TBSV RNA replicates in association with peroxisomes and TBSV-BS3-derived dsRNA is not a substrate for DCL2. The finding is interesting and the established system must be useful for future studies. Overall, the manuscript is written well. I list my comments and questions below.
1. Lines 44-45: Move ‘(RNAi)’ after ‘interference’.
2. Figures 1B, 3A, 3B, 5A, 5B, 5C, 5D, 6A, 6B, 7A, 7C, 7E, and Supplementary Figure 1: Indicate the positions of size markers (like in Fig. 3C) and/or positions of major RNA bands (TBSV genomic or subgenomic RNA, etc.). Are the markers in Fig, 7B (3 kb and 4 kb) dsRNA? Please describe it in the legend.
3. Lines 286-288, Fig. 1: ‘Blue line’ is not found.
4. Line 292: D electron -> Electron
5. Lines 292-317, Fig. 2: Evidence for the peroxisome localization of TBSV replication complex (RC) is rather weak. I found that the Fig. 8G data in reference 46 show that TBSV RC is formed on peroxisome in infected N. benthamiana cells. However, this is in N. benthamiana and the strain of TBSV is not described. I suggest the authors to show the localization of B2-GFP and a fluorescent protein-tagged peroxisome marker (e.g., RFP-SKL) in TBSV-BS3-infected A. thaliana cells.
Figure 2E and Supplementary Figure 3B: Show control images from uninfected cells.
Subcellular localization of TBSV-BS3 RC in N. benthamiana: In Supplementary Figure 3B, spherules seem to be formed on different types of organellar membranes (round vs irregular in shape, different pattern of staining; maybe indicated by white and black arrows). Are they all derived from peroxisomes?
6. Figure 3B: The dsRNA band patterns for total RNA from TBSV-infected and GFP-expressing or B2:GFP-expressing, or immunoprecipitated RNA from TBSV-infected and B2:GFP-expressing plants are different from each other. It would be useful if some more descriptions about the identities of each RNA bands could be described.
I presume the uppermost dsRNA band in the ‘total’ fractions represent the full-length TBSV dsRNA (so called replicative form, RF). If this is correct, the RF RNA is not found in the ‘GFP IP’ fraction. Together with the later results that some dsRNA species are detected only in the absence of DCL4 (Fig. 6B) and the generation of TBSV siRNA depends on DCL4 (Fig. 6C), I suspect that B2:GFP is not associated with replication complex but B2-stabilized dsRNA intermediate for viral secondary siRNA production. Have you examined the effect of sgs3 or rdr6 mutations on B2:GFP subcellular localization and/or the pattern of B2:GFP-associated dsRNA? Or have you examined the TBSV RNA-dependent RNA polymerase activity of GFP-immunoprecipitated fractions.
7. Fig. 5A, upper panel: Black and white (positive and negative) reversed?
8. Line 476, Fig. 6C: Loading control is missing.
9. Lines 496-497: An author’s memo remains to be removed.
10. I could not find legends to Supplementary Figures. Add them.
11. Supplementary Figures 5, 6 -> 4, 5?
Author Response
Point-by-point response to the comments by Reviewer 3
Reviewer comments are in italics, our responses in regular type.
Tomato bushy stunt virus (TBSV) is a positive-strand RNA virus for which abundant knowledge about RNA replication mechanisms has been obtained by using Saccharomyces cerevisiae as a model host. Although some studies on TBSV-plant interactions have been performed using a plant Nicotiana benthamiana, there was no report about TBSV infection in Arabidopsis thaliana, the most well-established model plant for genetic analyses. In this study, the authors found that the BS3 strain of TBSV is highly infectious on A. thaliana, and using this pathosystem, they show that TBSV RNA replicates in association with peroxisomes and TBSV-BS3-derived dsRNA is not a substrate for DCL2. The finding is interesting and the established system must be useful for future studies. Overall, the manuscript is written well. I list my comments and questions below.
Lines 44-45: Move ‘(RNAi)’ after ‘interference’.
Thank you, this has been corrected (Line 45).
Figures 1B, 3A, 3B, 5A, 5B, 5C, 5D, 6A, 6B, 7A, 7C, 7E, and Supplementary Figure 1: Indicate the positions of size markers (like in Fig. 3C) and/or positions of major RNA bands (TBSV genomic or subgenomic RNA, etc.). Are the markers in Fig, 7B (3 kb and 4 kb) dsRNA? Please describe it in the legend.
Size markers have been added where applicable. Concerning the size markers on the dsRNA northwestern blots, we need to specify why we could not always indicate the size. We used bacteriophage Phi6 dsRNA as a size marker (Thermo Scientific, now discontinued) on every north-western blot we performed. However, we found that while the Phi6 RNAs could easily be detected in agarose gel by EtBr staining, the corresponding bands were not always revealed upon B2 probing in northwestern blotting as previously reported in Monsion et al. (2018), see for example Fig 2E . This puzzling phenomenon may reflect partial denaturation of dsRNA species. Therefore, since in the blot shown in Figure 7B the Phi6 bands were visible, we included the size markers. These can be used to approximately compare this band pattern with the ones in the other figures, where the Phi6 bands could not be seen by northwestern blot. We would like to reiterate that this kind of blotting does not for the moment allow to determine the identity of the dsRNA species detected.
Lines 284-286, Fig. 1: ‘Blue line’ is not found.
Indeed. Figure 1 has been updated with the corresponding blue lines indicating probes.
Line 290: D electron -> Electron
We believe this is an artifact of “3D” from “3D electron” being recognized as a paragraph number by one of the softwares involved in processing/converting the manuscript. We are aware of the problem, and will keep an eye on it.
Lines 291-314, Fig. 2: Evidence for the peroxisome localization of TBSV replication complex (RC) is rather weak. I found that the Fig. 8G data in reference 46 show that TBSV RC is formed on peroxisome in infected N. benthamiana cells. However, this is in N. benthamiana and the strain of TBSV is not described. I suggest the authors to show the localization of B2-GFP and a fluorescent protein-tagged peroxisome marker (e.g., RFP-SKL) in TBSV-BS3-infected A. thaliana cells.
We agree that this would provide interesting information and strengthen the other results shown in the paper. However, transient expression in Arabidopsis upon agroinfiltration followed by virus infection is not as trivial as in N. benthamiana and generating B2:GFP lines expressing RFP:SKL would be a question of several months to say the least, so it is out of the scope of this submission. We believe that the electron microscopy on Arabidopsis shown here together with the localization of RFP:SKL-labeled peroxisomes in B2:GFP expressing N. benthamiana lines upon infection with TBSV-BS3 is sufficient evidence to claim peroxisomal replication of TBSV in Arabidopsis.
Figure 2E and Supplementary Figure 3B: Show control images from uninfected cells.
We have included in supplementary figure 3, electron microscopy micrographs of a peroxisome from an uninfected B2:GFP N. benthamiana. Unfortunately, given the short delay given for revision, we are unable to provide pictures of a peroxisome from uninfected B2:GFP A. thaliana.
Subcellular localization of TBSV-BS3 RC in N. benthamiana: In Supplementary Figure 3B, spherules seem to be formed on different types of organellar membranes (round vs irregular in shape, different pattern of staining; maybe indicated by white and black arrows). Are they all derived from peroxisomes?
This observation is correct. In Supplementary Figure 3B, while most spherules are found in association with round-shaped peroxisomal membranes, others are found on irregularly shaped membranes that seem to correspond to vacuolar membranes. Confirmation of this latter localization that includes the use of specific vacuolar membrane markers would require further investigations. We believe this is out of the scope of this manuscript.
The dsRNA band patterns for total RNA from TBSV-infected and GFP-expressing or B2:GFP-expressing, or immunoprecipitated RNA from TBSV-infected and B2:GFP-expressing plants are different from each other. It would be useful if some more descriptions about the identities of each RNA bands could be described.
Indeed, we would like to be able to determine the identities of the dsRNA bands to assess the changes caused by B2 on dsRNA populations. However, as stated above and in the manuscript, the northwestern blot technique for the moment does not provide any sequence information.
I presume the uppermost dsRNA band in the ‘total’ fractions represent the full-length TBSV dsRNA (so called replicative form, RF). If this is correct, the RF RNA is not found in the ‘GFP IP’ fraction. Together with the later results that some dsRNA species are detected only in the absence of DCL4 (Fig. 6B) and the generation of TBSV siRNA depends on DCL4 (Fig. 6C), I suspect that B2:GFP is not associated with replication complex but B2-stabilized dsRNA intermediate for viral secondary siRNA production. Have you examined the effect of sgs3 or rdr6 mutations on B2:GFP subcellular localization and/or the pattern of B2:GFP-associated dsRNA? Or have you examined the TBSV RNA-dependent RNA polymerase activity of GFP-immunoprecipitated fractions.
These are very interesting points and suggestions made by this reviewer, which we have wondered about ourselves. Concerning the presence in the B2 IPed fractions of dsRNA species smaller than the full-length replicative form, we attribute this phenomenon to technical limitations during IP, do to either RNA degradation in the lysates or retention by the beads of dsRNA of smaller molecular weight. It may also be due to the capture of RNAs that are “mid-replication” and therefore only partially double-stranded. In general, the presence of TBSV replicase in the B2:GFP-IPed fractions attests to the presence of replicating RNA.
As a dsRNA-binding protein, B2:GFP probably also binds host RDR-dependent dsRNA. As this reviewer probably is aware of, precise activity of RDR proteins in terms of when, how and where it acts on the viral RNA substrate is very poorly understood. Given the likelihood that RDR6 products are short-lived and rapidly processed by Dicer, such questions should be addressed comparing the RNA profiles B2 IPs from rdr6/dcl4/B2:GFP plants and dcl4/B2:GFP plants, for instance. While this would break considerable new ground in the field and we would be very much interested in doing it in the future, we argue this is out of the scope of this study. As this reviewer suggests, the additional small molecular weight dsRNA detected could be a product of a host RDR that is stabilized by dcl4 knock-out. A sentence including this hypothesis has been added to the discussion (lines 553-554). For this reviewer’s information, we have never detected any RDR protein peptides in our B2 IPs from healthy, TRV- and TBSV-infected plants. We interpret this as either (i) B2 directly or indirectly inhibiting RDR processing of viral RNA or (ii) RDR interaction with viral dsRNA being too transient to be detected by this approach. Concerning the last question, we have not tested the RNA-dependent RNA polymerase activity of GFP-immunoprecipitated fractions. It would also be very interesting to do, but outside the range of this paper.
Fig. 5A, upper panel: Black and white (positive and negative) reversed?
Thank you, this has been rectified.
Line 481, Fig. 6C: Loading control is missing.
We apologize. It has been added.
Lines 496-497: An author’s memo remains to be removed.
We apologize. There was confusion on the manuscript version submitted.
I could not find legends to Supplementary Figures. Add them.
Done.
Supplementary Figures 5, 6 -> 4, 5?
Thank you, this has been rectified.
Reference:
Monsion, B., Incarbone, M., Hleibieh, K., Poignavent, V., Ghannam, A., Dunoyer, P., Daeffler, L., Tilsner, J., and Ritzenthaler, C. (2018). Efficient Detection of Long dsRNA in Vitro and in Vivo Using the dsRNA Binding Domain from FHV B2 Protein. Front Plant Sci 9, 70.

Round 2
Reviewer 2 Report
- The authors’ stance regarding the role of DRB4 in TBSV derived vsiRNA production is still not clear for me. In their Author’s Notes they claim that the role of DRB4 in antiviral responses is highly variable in vivo, which explains the lack of effect on vsiRNA production in drb4 mutants as shown in Figure 5. In the manuscript however they state, quote: “decrease in TBSV-derived siRNA was observed in lines defective in DRB4 function (drb4, Figure 5D), consistently with the known role of DRB4 as a player in antiviral siRNA biogenesis and a co-factor of DCL4.” This point is especially critical since according to their confocal microscopy data DRB4 robustly colocalize with TBSV VRCs. On the other hand, transient over-expression of DRB4-tRFP (unlike DRB2-tRFP) has a negligible effect on TBSV replication. Combined, these observations imply that DRB4 may play a role in TBSV infection, however this is largely independent of DCL4 activity. Does this imply some alternative DCL4-independent role of DRB4 in TBSV replication? They should decide, which standpoint they adopt and should discuss this in the manuscript. Since the authors’ claim that one of the main points of their paper is the characterization of the non-canonical RNAi response to TBSV, this issue is not trivial.
- The references are not correctly cited in the discussion section. The Fatyol et al. paper (Journal of Virology vol. 94 Issue 11 e00017-20) is missing from the reference list. Additionally, the results of that paper are not correctly interpreted by the authors. The mentioned increased PVX accumulation was only observed upon transient DRB2-overexpression, which is the consequence of the inhibition of DCL activity and may not be physiologically relevant. More importantly however, the Fatyol et al paper reports that DRB2-deficiency led to increased accumulation of PVX during bona fide systemic virus infection. This result actually agrees quite well with the data presented by the Incarbone et al. paper. This section of the manuscript should be corrected.
- In line 566 TBSV should be changed to TRV to make sense of the sentence (“In these plants, TBSV (TRV?) infection leads to production of DCL2-dependent 22nt vsiRNA, as opposed to wild-type plants where vsiRNA are produced mainly by DCL4, suggesting an inhibitory activity of B2:GFP on DCL4 [46].”).
Author Response
Reviewer 2
The authors’ stance regarding the role of DRB4 in TBSV derived vsiRNA production is still not clear for me. In their Author’s Notes they claim that the role of DRB4 in antiviral responses is highly variable in vivo, which explains the lack of effect on vsiRNA production in drb4 mutants as shown in Figure 5. In the manuscript however they state, quote: “decrease in TBSV-derived siRNA was observed in lines defective in DRB4 function (drb4, Figure 5D), consistently with the known role of DRB4 as a player in antiviral siRNA biogenesis and a co-factor of DCL4.” This point is especially critical since according to their confocal microscopy data DRB4 robustly colocalize with TBSV VRCs. On the other hand, transient over-expression of DRB4-tRFP (unlike DRB2-tRFP) has a negligible effect on TBSV replication. Combined, these observations imply that DRB4 may play a role in TBSV infection, however this is largely independent of DCL4 activity. Does this imply some alternative DCL4-independent role of DRB4 in TBSV replication? They should decide, which standpoint they adopt and should discuss this in the manuscript. Since the authors’ claim that one of the main points of their paper is the characterization of the non-canonical RNAi response to TBSV, this issue is not trivial.
To address this point, we have added the following paragraph to the discussion (Lines 597-604):
« Our observations that DRB4 is not strictly necessary for DCL4-dependent production of TBSV-derived siRNA is in agreement with published studies on TuMV and TCV (Curtin et al, 2008; Qu et al., 2008). However, DRB4 has been shown to be mandatory for DCL4 activity during infection by TSWV and TYMV (Curtin et al, 2008; Jakubiec et al., 2012), suggesting that DRB4-DCL4 interplay is highly dependent on virus species. Interestingly, the moderate increase in TBSV accumulation observed in drb4 mutants is reminiscent of that observed in dcl4 mutants, suggesting that either (i) the vsiRNA produced by DCL4 in the absence of DRB4 are not functional in mediating antiviral RNAi, or (ii) DRB4 plays an antiviral role independently of DCL4, or both. »
Concerning the lack of effect of DRB4 over-expression on TBSV accumulation in N. benthamiana, despite the robust co-localization, we do not find this particularly problematic or surprising for the overall interpretation. One possibility is that Arabidopsis DRB4 binds viral dsRNA but does not recruit/interact with any N. benthamiana proteins. Another is that the knock-out of DRB4 has an effect on TBSV accumulation, while its over-expression does not, for example due to the saturation of the available pool of putative DRB4 co-factors (such as DCL4). Of course, the RNAi-suppression activity of B2 in the N. benthamiana could also play a role in these observations. Given all these possibilities, we prefer not to further speculate about the over-expression results.
We can for the moment only speculate on why over-expression of DRB2 has such dramatic effects on virus accumulation, while DRB4 does not. We think that the antiviral activities of these two DRB proteins may rely on very different mechanisms.
The references are not correctly cited in the discussion section. The Fatyol et al. paper (Journal of Virology vol. 94 Issue 11 e00017-20) is missing from the reference list. Additionally, the results of that paper are not correctly interpreted by the authors. The mentioned increased PVX accumulation was only observed upon transient DRB2-overexpression, which is the consequence of the inhibition of DCL activity and may not be physiologically relevant. More importantly however, the Fatyol et al paper reports that DRB2-deficiency led to increased accumulation of PVX during bona fide systemic virus infection. This result actually agrees quite well with the data presented by the Incarbone et al. paper. This section of the manuscript should be corrected.
We thank this reviewer for bringing this up. Put simply, we referenced the transient DRB2 overexpression experiment because it most resembled the experiment we show in Figure 5 and in our previous pre-print (also Incarbone et al., see references), and was therefore most comparable. We do not mention the effect of DRB2 overexpression on DCL activity (as observed in Fatyol et al) because in our manuscript we provide no data on this. The drastic decrease in TBSV gRNA and the presence of B2GFP would render small RNA blots on the N. benthamiana samples in Figure 5A of little utility.
We agree with this reviewer that the results obtained through VIGS of DRB2 by Fatyol et al should be added to the discussion. This has been done (Lines 593-597).
The reference in the bibliography was somehow lost in conversion/translation… We apologize, it has been implemented.
In line 566 TBSV should be changed to TRV to make sense of the sentence (“In these plants, TBSV (TRV?) infection leads to production of DCL2-dependent 22nt vsiRNA, as opposed to wild-type plants where vsiRNA are produced mainly by DCL4, suggesting an inhibitory activity of B2:GFP on DCL4 [46].”).
Indeed, this sentence does not express what we wanted to say, probably as a result of several rounds of editing. We thank this reviewer for the keen eye, and have changed the sentence accordingly to:
« The inability of DCL2 to generate 22nt TBSV vsiRNA is in agreement with the absence of vsiRNA in TBSV-infected 35S:B2:GFP/Col-0 plants. In these plants, we have previously shown that infection by TRV leads to production of DCL2-dependent 22nt vsiRNA, as opposed to wild-type plants where vsiRNA are produced mainly by DCL4, suggesting an inhibitory activity of B2:GFP on DCL4 »
Reviewer 3 Report
The authors have addressed my previous concern and appropriately revised the manuscript. I have only one minor comment about the legend to Supplementary Figure 3:
Lines 13-14 should be ‘(A) Confocal microscopy imaging of leaf epidermal cells from TBSV-infected 35S B2:GFP N. benthamiana at 3 dpi at low (left) and high magnification (right).’, or something like that.
Add explanation for panel F (I presume that these images are from uninfected B2:GFP N. benthamiana).
Author Response
The authors have addressed my previous concern and appropriately revised the manuscript. I have only one minor comment about the legend to Supplementary Figure 3:
Lines 13-14 should be ‘(A) Confocal microscopy imaging of leaf epidermal cells from TBSV-infected 35S B2:GFP N. benthamiana at 3 dpi at low (left) and high magnification (right).’, or something like that.
Thank you, this is corrected.
Add explanation for panel F (I presume that these images are from uninfected B2:GFP N. benthamiana).
This is corrected.